# Synthesis of a Novel Phosphorous-Nitrogen Based Charring Agent and Its Application in Flame-retardant HDPE/IFR Composites

**DOI:** 10.3390/polym11061062

**Published:** 2019-06-19

**Authors:** Junlei Chen, Jihui Wang, Aiqing Ni, Hongda Chen, Penglong Shen

**Affiliations:** 1School of Materials Science and Engineering, Wuhan University of Technology, 122 Luoshi Road, Wuhan 430070, China; junleichen@whut.edu.cn (J.C.); jhwang@whut.edu.cn (J.W.); Hongdachen@whut.edu.cn (H.C.); spl666@whut.edu.cn (P.S.); 2State Key Laboratory of Advanced Technology for Materials Synthesis and Processing, Wuhan University of Technology, 122 Luoshi Road, Wuhan 430070, China

**Keywords:** intumescent flame retardant, polyethylene composites, mechanical properties, thermal properties, flame retardancy mechanism

## Abstract

In this work, a novel phosphorous–nitrogen based charring agent named poly(1,3-diaminopropane-1,3,5-triazine-o-bicyclic pentaerythritol phosphate) (PDTBP) was synthesized and used to improve the flame retardancy of high-density polyethylene (HDPE) together with ammonium polyphosphate (APP). The results of Fourier transform infrared spectroscopy (FTIR) and ^13^C solid-state nuclear magnetic resonance (NMR) showed that PDTBP was successfully synthesized. Compared with the traditional intumescent flame retardant (IFR) system contained APP and pentaerythritol (PER), the novel IFR system (APP/PDTBP, weight ratio of 2:1) could significantly promote the flame retardancy, water resistance, and thermal stability of HDPE. The HDPE/APP/PDTBP composites (PE3) could achieve a UL-94 V-0 rating with LOI value of 30.8%, and had a lower migration percentage (2.2%). However, the HDPE/APP/PER composites (PE5) had the highest migration percentage (4.7%), lower LOI value of 23.9%, and could only achieve a UL-94 V-1 rating. Besides, the peak of heat release rate (PHRR), total heat release (THR), and fire hazard value of PE3 were markedly decreased compared to PE5. PE3 had higher tensile strength and flexural strength of 16.27 ± 0.42 MPa and 32.03 ± 0.59 MPa, respectively. Furthermore, the possible flame-retardant mechanism of the APP/PDTBP IFR system indicated that compact and continuous intumescent char layer would be formed during burning, thus inhibiting the degradation of substrate material and improving the thermal stability of HDPE.

## 1. Introduction

Polyethylene (PE) is a kind of thermoplastic material with light weight, non-toxic, excellent electrical insulation, chemical corrosion resistance, low cost, and easy processing. Therefore, it has a wide application in the fields of electrical appliances, chemical, food, machinery, and other industries [1,2,3]. Despite its many advantages, its application is limited due to its low limiting oxygen index (LOI), only about 18%, which belongs to flammable materials. PE is a non-carbonized polymer, which could burn easily because it has only carbon and hydrogen in its molecular chain. Almost all the long chains of PE are cracked into combustible gas during combustion, and there is no residual carbon residue.

Halogen-containing flame retardants have the characteristics of low cost, high flame retardant efficiency [4,5], and good compatibility, and occupy a large market in the current flame retardant of polypropylene. However, its combustion will produce toxic gas, pollute the environment, and cause secondary damage to the human body, which is prohibited in many fields [6]. Halogen-free flame retardants with low smoke emission, non-corrosive gas generation, and low toxicity have attracted extensive attention from researchers [7,8,9]. Intumescent flame retardant (IFR) is a halogen-free flame retardant with P, C, and N as the main core, including acid source, carbon source, and gas source [10]. When the polymer containing IFR is heated, a uniform carbon foam layer can be generated on the surface, which plays the role of heat insulation, oxygen insulation, smoke suppression, and prevents droplet phenomenon, so it has good flame retardant performance [11,12,13,14]. The key to the design of intumescent flame retardant is to form a dense and continuous fluffy carbon layer. The decomposition temperature of gas source should be compatible with that of charring agent and dehydrating agent. If the decomposition temperature is too low, the gas will be released before the formation of the carbon layer, which will not make the carbon layer fluffy; if the decomposition temperature is too high, the carbon layer will be pushed up and blown out.

At present, the most widely used IFR is mainly composed of ammonium polyphosphate (APP), pentaerythritol (PER), and melamine (MEL). But this traditional IFR system has many shortcomings of poor thermal stability and poor compatibility with the matrix due to the low molecular weight of PER and MEL, which deteriorate the flame retardancy and mechanical properties of the material [15,16]. Moreover, such as PER and MEL are water-soluble and moisture sensitive, which cause the flame retardant easily to be attacked by water and exuded out of the samples, leading to a worsening of the flame retardancy [17,18]. In view of the above problems, many scholars have carried out active research on the synthesis and application of macromolecule IFR, and have made phased progress. Li et al. [19] synthesized a novel charring agent with cyanuric chloride, ethanolamine, and ethylenediamine as main raw materials. It was mixed with ammonium polyphosphate to prepare flame retardant polypropylene (PP), and the thermal stability and flame retardancy of PP were obviously improved. Zhan et al. [20] synthesized a spiropentaerythritol diphosphate melamine (SPDPM) IFR, which can significantly improve the thermal stability and flame retardancy of polylactic acid (PLA). Liu et al. [21] synthesized a triazine oligomer charring agent (CA) from cyanuric chloride, ammonia water, and diethylenetriamine, and used it in flame retardant long glass fiber reinforced polypropylene (LGFPP) with APP and organic modified montmorillonite (OMMT). It was found that when the content of APP/CA/OMMT was 20 wt %, the LOI of the composites reached 31.3%. The vertical burning test achieved a UL-94 V-0 rating, and the carbon residue increased significantly at high temperature. Xie et al. [22] synthesized a novel hindered amine phosphorous–nitrogen macromolecular charring agent (HAPN) with free-radical quenching capability and mixed it with APP to flame-retard PP. When the content of HANP/APP was 25 wt %, PP/HAPN/APP passed the UL-94 V-0 test and the oxygen index could reach 29.5%. The above researches showed that triazine-derived macromolecule charring agent is an effective method to solve many shortcomings of traditional IFR, which could remarkably improve thermal stability and flame retardancy.

In this paper, a triazine-derived macromolecular charring agent named poly(1,3-diaminopropane-1,3,5-triazine-*o*-bicyclic pentaerythritol phosphate) (PDTBP) was synthesized and utilized to improve the flame retardant property of high-density polyethylene (HDPE) combined with APP by melt blending method. The chemical structure of PDTBP was characterized by Fourier transform infrared spectroscopy (FTIR) and ^13^C solid-state nuclear magnetic resonance (NMR). The flame retardancy, water resistance, mechanical properties, thermal stability, and flame-retardant mechanism of HDPE and HDPE/IFR composites were investigated by LOI, vertical burning test (UL-94), cone calorimetric test (CCT), tensile and flexural strength tests, thermogravimetric analysis (TGA), TG-FTIR, scanning electron microscopy (SEM), and Raman spectroscopy.

## 2. Experiments

### 2.1. Materials

HDPE (HMA 028, melt flow index of 40 g/10 min (190 °C, 2.16 Kg), density of 0.954 g/cm^3^) was supplied by Exxon Mobil Corporation, Avon, Texas, USA. APP (polymerization degree ≥1000) was purchased from Jinan Sennuo New Material Technology Co., Ltd., Jinan, China. PER, acetonitrile, and triethylamine were obtained from Sinopharm Chemical Reagent Co., Ltd., Shanghai, China. Cyanuric chloride (CNC) and diaminopropane were obtained from Shanghai Aladdin Bio-Chem Technology Co., Ltd., Shanghai, China. 2,6,7-Trioxa-1-phosphabicyclo-[2,2,2]octane-4-methanol-1-oxide (PEPA) was supplied from Guangzhou Xijia Chemical Co., Ltd., Guangzhou, China. All the commercial materials were used without further purification.

### 2.2. Synthesis of PDTBP

Firstly, 18.45 g (0.1 mol) CNC and 100 mL acetonitrile were added in a 500 mL round-bottom flask and stirred until a transparent solution was formed. Next, 18.00 g (0.1 mol) PEPA was dissolved into another 100 mL acetonitrile. Then, the mixture and 13.9 mL (0.1 mol) triethylamine were added dropwise into the CNC solution over 1 h and the reaction lasted for 3 h at room temperature. Afterwards, a solution of 4.2 mL (0.05 mol) diaminopropane and 13.9 mL (0.1 mol) triethylamine in 30 mL acetonitrile was added dropwise into the flask over 1 h and the reaction lasted for another 3 h at 45–50 °C. Thereafter, another solution of 4.2 mL (0.05 mol) diaminopropane and 13.9 mL (0.1 mol) triethylamine in 30 mL acetonitrile was added dropwise into the flask. The mixture was heated to 90–100 °C with refluxing for 6 h. The precipitate was then filtered and washed several times with distilled water and anhydrous ethanol. Finally, the products were dried under a vacuum at 80 °C for 24 h and the novel charring agent PDTBP was obtained. The synthesis route is shown in Scheme 1.

### 2.3. Preparation of Flame-Retardant HDPE/IFR Composites

The HDPE, APP, PDTBP, and PER were dried in a vacuum oven at 80 °C for 10 h before use. Then HDPE and flame retardants were mixed for 10 min through a two-roll mixing mill (Rheomixer XSS-300, Shanghai Ke Chuang China) at 180 °C and 60 rpm. The mixed composites were molded under compression (10 MPa) at 180 °C for 10 min and cooled to room temperature to obtain HDPE/IFR composite sheets with standard size for tests. The formulations of the flame-retardant HDPE/IFR composites are listed in Table 1.

### 2.4. Characterization

FTIR was measured by using a Nexus infrared spectrometer (Thermo Nicolet, Waltham, MA, USA), and the measurement was carried out in the optical range of 4000–400 cm^−1^.

The ^13^C solid-state NMR spectrum were recorded on an Avance III HD 400 MHz spectrometer (Bruker Inc., Bremen, Germany).

The LOI values were determined by an oxygen index meter (JF-3, Jiangning Analysis Instrument Factory, Nanjing, China) according to ASTM D2863-2008. The dimensions of the specimens were 130 mm × 6.5 mm × 3.0 mm.

The vertical burning test (UL-94) was conducted on a vertical burn instrument (PX-03-001, Phinix Analysis Instrument Co., Ltd., Suzhou, China) according to ASTM D3801-2010. The dimensions of the specimens were 125 mm × 13 mm × 3.0 mm.

Water resistance test: The samples for LOI and UL-94 vertical burning tests were put in distilled water at 70 °C for 168 h, and the water was replaced every 24 h. The treated samples were subsequently taken out and dried under a vacuum at 80 °C to a constant weight. The weight of the sample for LOI test was measured before water soaking (W1) and after drying (W2). The migration percentage was calculated as the following equation: (W1-W2)/W1 × 100%. The test was operated five times and the average value was reported. Moreover, the LOI value and UL-94 vertical rating of the composites after water soaking were also determined.

The CCT was conducted on a cone calorimeter (Fire Testing Technology Co., East Grinstead, UK) according to ISO 5660. The dimensions of the specimens were 100 mm × 100 mm × 3.0 mm, and each specimen was wrapped in aluminum foil and exposed horizontally to an external heat flux of 35 kW/m^2^. The residues of the specimens after the test were photographed by a digital camera (DSC-RX10 II, SONY Inc., Tokyo, Japan).

The tensile strength and flexural strength of HDPE and HDPE/IFR composites were measured on an Instron 5967 universal testing machine (Instron Corporation, High Wycombe, UK), according to ASTM D638 (crosshead speed 10 mm/min) and ASTM D790 (in a three-point loading mode), respectively. All samples were tested five times and the average value was reported.

The TGA test was carried out with a STA449F3 thermal analyzer (Netzsch Instruments Co., Selb, Germany), and the test temperature ranged from 100 °C to 800 °C (10 mg sample, 10 °C /min heating rate, nitrogen and air atmosphere).

The TG-FTIR instrument consists of a thermogravimeter (STA449F3, Netzsch Instruments Co., Selb, Germany) and a Fourier transform infrared spectrometer (Thermo Nicolet 6700, Waltham, MA, USA). The investigation was carried out from room temperature to 900 °C at a heating rate of 10 °C /min under a nitrogen flow of 30 mL/min.

The SEM (JSM-IT300, JEOL Ltd., Tokyo, Japan) was utilized to observe the morphology of the residual char, and Raman spectrum was conducted on a Confocal Raman Microprobe (Invia, Renishaw Co., London, UK) using a 633 nm helium–neon laser at room temperature. 

## 3. Results and Discussion

### 3.1. Characterization of PDTBP

The FTIR spectra of PEPA and PDTBP are shown in Figure 1. As for the PEPA, an O-H band at 3391 cm^−1^, C-H bands at 2964 and 2910 cm^−1^, a P=O band at 1297 cm^−1^, a P-O-C band at 1019 cm^−1^, and a CH_2_-OH band at 957 cm^−1^ could be observed [23,24]. While the spectrum of PDTBP mainly presented the following peaks: 3416 and 3259 cm^−1^ (N-H), 2939 and 2869 cm^−1^ (C-H of -CH_2_-), 1576 cm^−1^ (C=N), 1361 cm^−1^ (tr-N, tr meant triazine ring), 1316 cm^−1^ (P=O), 1287 cm^−1^ (C-N), 1249 cm^−1^ (tr-O), 1099 cm^−1^ (C-O), 1057 and 985 cm^−1^ (P-O-C), and 799 cm^−1^ (tr) [25,26]. A broad peak (N-H) instead of a sharp peak (-OH) of PEPA around 3391 cm^−1^ and a disappeared peak at 850 cm^−1^ (C-Cl) of CNC [27] in the PDTBP spectrum indicated that PEPA and diaminopropane had been reacted with CNC.

The ^13^C solid-state NMR spectrum was used to further characterize the chemical structures of PDTBP. Figure 2 shows the ^13^C solid-state NMR spectra of PDTBP. The peak at 165.8 ppm corresponded to the carbon atoms (a) of the triazine ring [26,28]. The signals located at 66.6 ppm and 7.9 ppm were assigned to -CH_2_- groups (b) and -C(CH_4_) (f) in the side chain, respectively. The peak at 38.7 ppm was ascribed to the the -CH_2_- groups (c) of the caged phosphate moiety. The peaks at 30.5 ppm and 13.1 ppm were attributed to the C atoms of the -CH_2_-CH_2_-CH_2_- (d) and -NH-CH_2_- (e) groups. All the above analysis indicates that the target triazine-derived macromolecular charring agent PDTBP was successfully synthesized.

### 3.2. Flame Retardancy and Water Resistance of Flame-Retardant HDPE/IFR Composites

The LOI values and UL-94 vertical test results of HDPE and HDPE/IFR composites before water soaking are tabulated in Table 2. Photos of char residues after the LOI test and photos of the UL-94 test for HDPE and HDPE/IFR composites are presented in Figure 3 and Figure 4, respectively. Pure HDPE exhibited a low LOI value of only 18.3%, burned violently with drops falling, and had no rating in the UL-94 vertical test. When the IFR system (APP/PDTBP or APP/PER) was added into HDPE with 30 wt %, the LOI value of HDPE/IFR composites increased significantly. The LOI value of PE3 could be increased to 30.8%, and it could be quickly extinguished after ignition and achieved a UL-94 V-0 rating when the weight ratio of APP and PDTBP was 2:1. However, PE5 had a UL-94 V-1 rating with a LOI value of 23.9%. Compared with PE5 contained traditional IFR system (APP/PER), PE3 showed better flame retardancy. 

Besides, Table 2 also presents the migration percentages and changes in flame retardancy of the samples after water soaking in 70 °C water for 168 h. All the HDPE/IFR composites showed a decrease in LOI value due to the precipitation of flame retardant. Meanwhile, the HDPE/APP/PDTBP composites had a low migration percentage (2.1%, 2.2% and 2.8% for PE2, PE3, and PE4, respectively) and could still achieve a UL-94 V-0 rating for PE3, whereas the HDPE/APP/PER composite had the highest migration percentage (4.7%) and had no burning rating. These may be because PDTBP has a higher molecular weight than PER, therefore it presents lower solubility and better compatibility with HDPE [29]. Based on the above results, it can be concluded that there is an excellent synergistic effect between APP and PDTBP which can form an intumescent char layer (PE3 in Figure 3), and the novel IFR system (APP/PDTBP, weight ratio was 2:1) is more effective in improving flame retardancy of HDPE compared with traditional IFR.

The CCT was applied to further investigate the combustion behavior of flame-retardant polymer in a real fire environment [30]. Figure 5 displays the heat release rate (HRR) and total heat release (THR) curves of pure HDPE and flame-retardant HDPE/IFR composites, and Table 3 lists the correlative characteristic parameters of time to ignition (TTI), peak of HRR (PHRR), time to PHRR (TPHRR), THR, fire performance index (FPI=TTI/PHRR), and char residue. It clearly showed that pure HDPE (PE1) had a sharp peak appeared at 346 s with the PHRR as high as 603.6 kW/m^2^. However, the PHRR and THR of the HDPE/APP/PDTBP composite (PE3) and HDPE/APP/PER composite (PE5) were significantly decreased. The PHRR and THR of PE3 were 173.5 kW/m^2^ and 51.8 MJ/m^2^, which were 44.6% and 45.0% lower than those of PE5, respectively. This was mainly due to the formation of a thick intumescent char layer, which was further confirmed by char residue of PE3 after CCT. It indicated that the thick intumescent char layer of PE3 formed by APP/PDTBP IFR system could protect the substrate resin from burning by inhibiting the transfer of heat and combustible gases. Besides, FPI [31] could be used to evaluate the fire hazard, and the higher FPI value correspond to lower fire hazard [32]. PE3 possessed the highest FPI value, which indicated the fire hazard of PE3 was much lower than that of PE1 and PE5. Moreover, the char residue of PE3 was much higher than that of PE1 and PE5. All the above results show that the APP/PDTBP (weight ratio was 2:1) IFR system has excellent flame retardancy for HDPE, which is in good agreement with the results of LOI and UL-94 vertical burning test.

### 3.3. Mechanical Properties of Flame-Retardant HDPE/IFR Composites

Mechanical properties such as tensile strength and flexural strength of HDPE and HDPE/IFR composites are shown in Figure 6. The tensile strength and flexural strength of pure HDPE (PE1) were 20.60 ± 0.50 MPa and 23.13 ± 0.68 MPa, respectively. With the addition of IFR system, all HDPE/IFR composites showed a reduction in tensile strength compared with pure HDPE, indicating that IFR system had a negative effect on tensile property of HDPE composites. For APP/PDTBP IFR system (PE2-PE4), the tensile strength of HDPE/IFR composites slightly increased with increasing the addition of PDTBP due to the macromolecular chain of PDTBP [32], which could be retained at 79%–83%. Besides, HDPE/APP/PER composite (PE5) had the lowest mechanical properties than that of pure HDPE and HDPE/APP/PDTBP composites (PE1–PE4), which was mainly due to the low molecular weight of PER and the poor compatibility between the APP, PER, and HDPE matrix [33]. However, the flexural strength of HDPE/IFR composites (PE2–PE4) were improved compared with PE1 and PE5. The improved flexural strength of the composites may be attributed to the high stiffness of the phosphorus layer of APP [34] and the lower polarity of PDTBP than PER which improve the compatibility between IFR system and HDPE matrix [33]. In terms of flame retardant properties and mechanical properties of the HDPE/IFR composites, PE3 has the best comprehensive performance with UL-94 V-0 rating, LOI value of 30.8%, tensile strength and flexural strength of 16.27 ± 0.42 MPa and 32.03 ± 0.59 MPa, respectively.

### 3.4. Thermal Properties and TG-FTIR Analysis

Figure 7 displays the TGA curves of APP, PDTBP and IFR (APP/PDTBP, weight ratio was 2:1) as well as IFR calculation in nitrogen and air atmosphere, and the corresponding TGA data are listed in Table 4. For PDTBP, its initial decomposition temperatures (T_i_, the temperature at 5.0 wt % mass loss) were 372.7 °C and 355.9 °C in nitrogen and air atmosphere respectively, indicating that it had outstanding thermal stability and could be enough to meet the processing temperature of HDPE. Meanwhile, the char residue of IFR at 800 °C were as high as 52.1% and 40.5% in nitrogen and air atmosphere respectively, indicating that IFR exhibited excellent char-forming ability and could be used as an efficient flame retardant. The IFR calculation curves were calculated using the experimental results and percentages of APP and PDTBP in IFR system according to Equation (1) [35]. It is obvious that the experimental curve was lower than the IFR calculation curve before 660 °C and was higher than the IFR calculation curve after 660 °C, however, the char residue at 800 °C of IFR was much higher than that of the calculated curve both in nitrogen and air atmosphere. Furthemore, the T_i_ of IFR (338.2 °C in nitregen and 321.5 °C in air) was lower than the calculated value (353.2 °C in nitregen and 331.1 °C in air) due to phosphorylation, dehydration, and carbonization between APP and PDTBP at a lower temperature [36]. Compared with the TGA data in nitrogen atmosphere, it can be found that T_i_ and char residue in air were both lower than that in nitrogen. The lower T_i_ may be caused by the oxygen in air, and the lower char residue may be due to the oxidative degradation of the char residue in the high temperature range of 550–800 °C. In conclusion, synergistic effect was generated between APP and PDTBP in IFR system (Equation (1)).
(1)$$w_{calculation}=w_{APP}\times 66.7\%+w_{PDTBP}\times 33.3\%,$$

Figure 8 illustrates the TGA and derivative thermal gravimetric analysis (DTG) curves of HDPE and flame-retardant HDPE/IFR composites in nitrogen atmosphere, and the related data are presented in Table 5. For pure HDPE (PE1), the T_i_ and maximum weight loss rate temperature (T_max_) were 438.3 °C and 483.8 °C with almost no char residue left at 800 °C. Compared with pure HDPE and flame-retardant HDPE/IFR composites, the HDPE/APP/PER composite (PE5) showed the lowest T_i_ (316.3 °C) due to the low thermal stability of PER. When the APP/PDTBP IFR system was introduced into the HDPE matrix, composites showed a lower T_i_ because, with the IFR system, the thermal decomposition and crosslinking reactions occurred at a low temperature [37]. Consequently, the char residue at 800 °C of PE3 was 14.7%, higher than that of PE5 and PE contained other IFR [34,38]. This indicates that the IFR system (APP/PDTBP, weight ratio was 2:1) can form a protective char layer before HDPE degradation, which prevents the transmission of heat, flammable gas, and oxygen required for material combustion [8,19], thus inhibiting the degradation of matrix materials and improving the thermal stability of HDPE [39,40].

Figure 9 shows the TGA and DTG curves of HDPE and flame-retardant HDPE/IFR composites in air atmosphere, and the corresponding data are on listed in Table 5. Compared with the TGA and DTG results in nitrogen atmosphere (Figure 8), the T_i_ and char residue in air were both decreased due to stimulating action of oxygen in air. Moreover, it is clear that HDPE/IFR composites have two decomposition stage in air atmosphere. The first decomposition stage was slightly shifted to low temperatures whereas the second decomposition stage was found in the high temperature range of 550–800 °C because of the oxidative degradation of the char residues in the hot air, leading to a lower char residue than that in nitrogen. The char residue formed in the first decomposition stage can act as barrier to enhance thermal stability of HDPE/IFR composites at high temperature. It is obviously found that PE3 has the highest char residue (8.9%), which indicates that its char residue presents the best resistance to thermal oxidation in the high temperature range.

In order to further explore the flame-retardant mechanism of the APP/PDTBP IFR system, the gaseous pyrolysis products of IFR produced during heating were analyzed by TG-FTIR. Figure 10 displays the FTIR spectra of pyrolysis products of IFR system (APP/PDTBP, weight ratio is 2:1) at different temperatures. It can be seen that there was almost no infrared absorption signal below 280 °C, indicating that the IFR system did not decompose below this temperature. Thus the IFR system was thermally stable for melting-blend with HDPE. With temperature increasing, the infrared absorption peaks at about 931 cm^−1^ and 966 cm^−1^ were detected, which were attributed to NH_3_ decomposed by APP at 350 °C. At 380 °C, the absorptions of H_2_O (3500-3800 cm^−1^ and 1625 cm^−1^) and CO_2_ at 2275-2395 cm^−1^ evolved during the carbonization process of IFR system [41]. Then when the temperature reached to 420 °C, the maximum signal intensity appeared, which meant that the maximum decomposition rate of IFR system occurred. Afterwards, the signal intensity decreased gradually indicating that the decomposition rate of IFR system was reduced by the char formation. The above results of gaseous pyrolysis products were quite consistent with the TGA of the IFR system in nitrogen atmosphere. The absorption signal of H_2_O and CO_2_ appeared in the same temperature range (380–550 °C) with the main release period of NH_3_, which indicated that the decomposition temperature of a gas source could be well compatible with that of charring agent, so that an intumescent char layer could be formed.

### 3.5. Characterization of Char Residues

The flame retardant efficiency depends largely on the quality of intumescent char layer, which is helpful to understand the flame retardant mechanism. Figure 11 shows the macrographs and SEM images of the char residues for PE3 and PE5 after CCT. Apparently, the char residue of PE5 had only a small expansion and the surface was rough and loose (Figure 11b,d), while the char layer of PE3 was highly expansive, continuous, and compact (Figure 11a,c). The micromorphologies of the char residues were also observed by SEM. It could be clearly seen that the char layer surface of PE5 (Figure 11f) was discontinuous with many holes. On the contrary, the char layer surface of PE3 (Figure 11e) was compact and continuous. This could effectively have a shielding effect and prevent the substrate resin from further burning.

Raman spectroscopy is a highly effective tool to analyze carbonaceous materials formed in the intumescent char [42]. Therefore, the Raman spectra of the char residues for PE3 and PE5 after CCT were tested to further investigate the structure of char residues, and Figure 12 presents the Raman spectrum of the char residues which were fitted into two Gaussian bands around 1380 cm^−1^ (D band) and 1600 cm^−1^ (G band). The microstructure of the residual char can be estimated by the ratio of the intensity of the D and G bands (I_D_/I_G_) [43], where I_D_ and I_G_ are the integrated intensities of the D and G bands, respectively. More importantly, the ratio of I_D_/I_G_ was inversely proportional to an in-plane microcrystalline size [44]. Besides, the smaller size of carbonaceous microstructures meant a better shield effect of char layer from combustion and better flame retardancy [45]. As shown in Figure 12, the I_D_/I_G_ ratio of PE3 (3.13) was greater than that of PE5 (2.83), which indicated that PE3 possessed a smaller carbonaceous microstructure and better flame retardant performance. Hence, the APP/PDTBP (weight ratio was 2:1) IFR system has excellent flame retardancy for HDPE compared with traditional flame retardants, which is consistent with the fire test results.

### 3.6. Flame-Retardant Mechanism

Based on the above analysis, the possible flame-retardant mechanism of the APP/PDTBP IFR system for HDPE is presented in Figure 13. During the combustion, PDTBP and APP had good synergistic effect in the flame retardant HDPE/IFR composites. At low temperature, the main decomposition products were triazine oligomers radicals [25], pentaerythriol, and phosphoric acid [46] decomposed from PDTBP, and oligomeric phosphate [22] produced by APP; meanwhile, the incombustible gases such as NH_3_ and H_2_O diluted the concentration of combustible gas and absorbed a lot of heat. As the temperature increases, these decompositon products formed a cross-linking structures [33]. Simultaneously, the NH_3_, H_2_O, and CO_2_ could make the system expand and foam, thus forming a compact and intumescent char layer, which could prevent the substrate material from further burning.

## 4. Conclusions

A novel charring agent PDTBP was synthesized and characterized. The APP/PDTBP IFR system could significantly promote the flame retardant properties, water resistance properties, and thermal stability of HDPE. When the weight ratio of APP and PDTBP was 2:1 with 30 wt % loading, the HDPE/IFR composites could have a LOI value of 30.8% and achieve a UL-94 V-0 rating both before and after water soaking, but the HDPE/APP/PER composite only had a LOI value of 23.9% and achieved a UL-94 V-1 rating before water soaking and no rating after water soaking. In addition, the HDPE/IFR (APP/PDTBP, weight ratio was 2:1) composite had lower PHRR, THR, fire hazard, and higher tensile strength (16.27 ± 0.42 MPa) and flexural strength (32.03 ± 0.59 MPa) than that of the HDPE/APP/PER composite. SEM, TG-FTIR, and Raman spectroscopy indicated that the IFR system (APP/PDTBP, weight ratio was 2:1) could form a compact and continuous intumescent char layer during the combustion, preventing the transfer of heat, flammable gas, and oxygen, thus effectively protected the HDPE matrix.

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
