# Peer review of "Synthesis of a Novel Phosphorous-Nitrogen Based Charring Agent and Its Application in Flame-retardant HDPE/IFR Composites"

_polymers, 2019, doi:10.3390/polym11061062_

Round 1

Reviewer 1 Report

Chen et.al. prepared the manuscript entitled “Synthesis of a Novel Phosphorous-Nitrogen Based Charring Agent Containing Caged Bicyclic Phosphate and Triazine Structure and Its Application in Flame- Retardant HDPE/IFR Composites” with detailed study. However, some issues should be solved before sending to the publication.

1.      The title should be a concise level and attractive type. Please modify the title of the manuscript.

2.      The results revealed in the abstract should be more quantitative with some comparisons. Should check the English grammar too. Please correct all sentences in the abstract and make it shorter sentences. For example, “The chemical structure of PDTBP was investigated by Fourier transform infrared spectroscopy (FTIR), (and??) 13C solid-state nuclear magnetic resonance (NMR)”, please check the sentence and correct it.

3.      Should give some updated references in the introduction part, for instance, the followed article can be cited at line no.39 in the manuscript.

“Polymer degradation and stability, 2019, 163, 185-194”

4.      The authors if possible, can mention in the experimental section “ the density, molecular weight, and melt flow index of HDPE” used in this study.   

5.      For water resistance test, the authors give details regarding approximately how much initial weight sample have taken to conduct the water resistance experiment.

6.      For tensile test experiment how many specimens tested to take average value?

7.      Line 286-289 should rewrite for better explanation.

8.      “Flame-Retardant Mechanism” interpretation should verified by some published papers.

9.      The temperature range mentioned in the figure contradicted with its characterization part.

10.  Water resistance test, the migration percentage part need more details information with separate paragraphs. Please check it.

11.  Figures are good but still need improvement. In Raman part, Line 317-319, Cite below paper. Cellulose (2013) 20:687–698. I think with [39], you can use this paper.

12.  Conclusion should be recheck. Make it more quantitative.

13.  Line 163-166, mentioned peaks need refernces.

Author Response

Point 1: The title should be a concise level and attractive type. Please modify the title of the manuscript.

Response 1: The title of the manuscript has been modified with a concise level and attractive type.

Point 2: The results revealed in the abstract should be more quantitative with some comparisons. Should check the English grammar too. Please correct all sentences in the abstract and make it shorter sentences. For example, “The chemical structure of PDTBP was investigated by Fourier transform infrared spectroscopy (FTIR), (and??) 13C solid-state nuclear magnetic resonance (NMR)”, please check the sentence and correct it.

Response 2: We have quantitatively analyzed the results with some comparisons, checked the English grammar, and corrected all sentences in the abstract with proper presentation.

Point 3: Should give some updated references in the introduction part, for instance, the followed article can be cited at line no.39 in the manuscript.

“Polymer degradation and stability, 2019, 163, 185-194”

Response 3This recommended reference of "Polymer degradation and stability, 2019, 163, 185-194" has been added in the introduction part.

Point 4: The authors if possible, can mention in the experimental section “the density, molecular weight, and melt flow index of HDPE” used in this study.

Response 4: The density and melt flow index of HDPE are mentioned in the experimental section, but detailed data on molecular weight are not available.

Point 5: For water resistance test, the authors give details regarding approximately how much initial weight sample have taken to conduct the water resistance experiment.

Response 5: The samples for LOI  and UL-94 vertical burning tests were used for water resistance test. The weight of the sample for LOI test was measured before water soaking and after drying for the migration percentage calculation, and the test was operated for five times and the average value was reported.

Point 6: For tensile test experiment how many specimens tested to take average value?

Response 6: Five specimens for tensile and flexural test were tested to take average value.

Point 7: Line 286-289 should rewrite for better explanation.

Response 7: Line 286-289 has been rewritten for better explanation.

Point 8: “Flame-Retardant Mechanism” interpretation should verified by some published papers.

Response 8: Several published papers have been cited to support the “Flame-Retardant Mechanism” interpretation.

Point 9: The temperature range mentioned in the figure contradicted with its characterization part.

Response 9: The temperature range mentioned in the figure and characterization part has been modified consistently.

Point 10: Water resistance test, the migration percentage part need more details information with separate paragraph. Please check it.

Response 10: Water resistance test, the migration percentage part has been described in more detail with separate paragraph.

Point 11: Figures are good but still need improvement. In Raman part, Line 317-319, Cite below paper. Cellulose (2013) 20:687–698. I think with [39], you can use this paper.

Response 11: In Raman part, the recommended paper of “Cellulose (2013) 20:687–698” has been cited.

Point 12: Conclusion should be recheck. Make it more quantitative.

Response 12: We have revised the conclusion part carefully and made it more quantitative.

Point 13: Line 163-166, mentioned peaks need refernces.

Response 13: Some references have been cited for the mentioned peaks.

Reviewer 2 Report

This paper is clearly written and easy to be read. The experimetal design is sound and characterization is carred out comprehensively.

I only recommand to the authors

to delete the word "could" in line 232

to put the main wavelength numbers in fig 1 (FTIR)

to better justify the need for water resistence of fire retarded HDPE

the better explain what they intend with  the sentence "smaller size of carbonaceous microstructure" (line 319-320)

Author Response

Point 1: This paper is clearly written and easy to be read. The experimental design is sound and characterization is carred out comprehensively.

I only recommand to the authors

to delete the word “could” in line 232

to put the main wavelength numbers in fig 1 (FTIR)

to better justify the need for water resistence of fire retarded HDPE

the better explain what they intend with the sentence “smaller size of carbonaceous microstructure” (line 319-320)  

Response 1: We have deleted the word could in line 232, put the main wavelengh numbers in fig. 1 (FTIR), and described the need for water resistence of fire retarded HDPE and the sentence smaller size of carbonaceous microstructure (line 319-320).

Reviewer 3 Report

English language and style  requires revision. The text is legible, but presents several grammatical errors, particularly on pages 7 e 8.

The TGA and DTG curves were performed only under nitrogen. They should also be made in the air. This would make more sense, since the compounds tested should be used as flame-retardants, obviously in the air.

No actual combustion tests are shown, only an evaluation table based on official parameters, and LOI determinations. These figures would be much more convincing if the prectical results of vertical and horizontal combustion tests were unequivocally demonstrated with snapshots, determining in the mean time  the minimum add-ons necessary in both types of experiment  to achieve self-extinguishing or even lack of ignition, if any.

Author Response

Point 1: English language and style requires revision. The text is legible, but presents several grammatical errors, particularly on pages 7 e 8.

The TGA and DTG curves were performed only under nitrogen. They should also be made in the air. This would make more sense, since the compounds tested should be used as flame-retardants, obviously in the air.

No actual combustion tests are shown, only an evaluation table based on official parameters, and LOI determinations. These figures would be much more convincing if the prectical results of vertical and horizontal combustion tests were unequivocally demonstrated with snapshots, determining in the mean time the minimum addtions necessary in both types of experiment to achieve self-extinguishing or even lack of ignition, if any.

Response 1: We have checked English language and style carefully, and corrected the grammatical errors.

The TGA test in the air has been added and the results have been analyzed in the manuscript.

Figures of  vertical combustion test have been added to the manuscript. In this paper, we aim to compare the advantages and disadvantages of novel IFR (APP/PDTBP) system and traditional IFR (APP/PER) system under the same amount of addition, and determine the best ratio of APP and PDTBP in novel IFR system. In the following study, we will change the amount of flame retardant to determine the minimum amount that can meet the flame retardant performance. We hope to get your understanding.

Round 2

Reviewer 1 Report

Accepted in present form.

Reviewer 3 Report

This document is now, in my opinion, worthy of being published